# YOUR DENSE RETRIEVER IS SECRETLY AN EXPEDITIOUS REASONER

## ABSTRACT

Dense retrievers enhance retrieval by encoding queries and documents into continuous vectors, but they often struggle with reasoning-intensive queries. Although Large Language Models (LLMs) can reformulate queries to capture complex reasoning, applying them universally incurs significant computational cost. In this work, we propose Adaptive Query Reasoning (AdaQR), a hybrid query rewriting framework. Within this framework, a Reasoner Router dynamically directs each query to either fast dense reasoning or deep LLM reasoning. The dense reasoning is achieved by the Dense Reasoner, which performs LLM-style reasoning directly in the embedding space, enabling a controllable trade-off between efficiency and accuracy. Experiments on large-scale retrieval benchmarks BRIGHT show that AdaQR reduces reasoning cost by 28% while preserving—or even improving—retrieval performance by 7%[1].

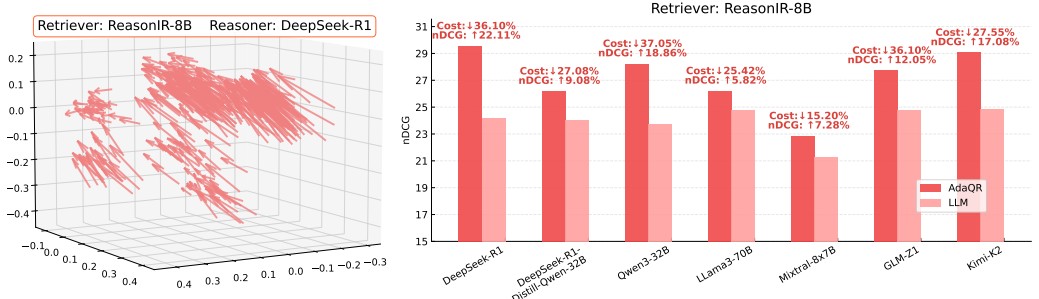

Figure 1: *Left*: PCA-reduced visualization of embedding transformation, where each arrow denotes the shift from the original query embedding to its reasoned counterpart, with most following systematic and structured trajectories. *Right*: AdaQR on the ReasonIR-8B dense retrievers yields substantial gains in retrieval performance and query rewriting efficiency on BRIGHT benchmark.

## 1 INTRODUCTION

Information Retrieval (IR) is a fundamental technology that bridges user queries and relevant documents across vast corpora. It plays a pivotal role in search engines, question answering system, etc (Bai et al., 2022; Muennighoff et al., 2023; Jin et al., 2025). Traditional approaches primarily rely on keyword matching (Robertson & Zaragoza, 2009) to evaluate relevance. While effective, these methods often struggle with capturing deeper semantics and contextual nuances (Chen et al., 2024a). Dense retrieval methods address these limitations by encoding queries and documents into continuous vector representations and performing similarity search in the embedding space. These approaches enable the retrieval system to handle contextually complex queries, and significantly improve recall performance (Zhao et al., 2024; Lin et al., 2025b; Zhang et al., 2025c).

However, for many reasoning-intensive real-world queries, the conventional embeddings produced by dense retrievers often fail to capture the relevance between a query and the retrieved documents—relevance that only becomes evident after further reasoning (Su et al., 2025; Chen et al.,

---

[1]Our code will be released to facilitate further research.

2025a). Recently, Large Language Models (LLMs), thanks to their increasingly powerful reasoning capabilities, have been employed to reformulate original queries, substantially enhancing performance in reasoning-intensive retrieval scenarios (Kostric & Balog, 2024; Dharwada et al., 2025). Nevertheless, applying LLM-based query rewriting to all queries in online or large-scale retrieval systems incurs substantial computational and latency costs, thereby becoming a bottleneck (Nguyen et al., 2025; Qin et al., 2025).

The primary cost of LLM reasoning arises from its auto-regressive and often lengthy generation process (Lin et al., 2025a). Consequently, a natural direction for optimization is to avoid performing this explicit reasoning step altogether. This raises an interesting question: *Is it possible for the reasoning process to be carried out implicitly?* Fortunately, an affirmative answer has been given to this question by recent latent reasoning studies, in which a model performs reasoning implicitly within its internal representations (Li et al., 2025; Chen et al., 2025b). However, these methods still rely on LLMs as the reasoner and can't meet the efficiency needs in high-throughput retrieval scenarios (Shen et al., 2025). A more efficient solution is to achieve this latent reasoning process directly by the dense retriever, which is feasible as we observe that: *the embeddings of some queries before and after LLM reasoning exhibit systematic, structured transformation* (see Figure 1, left). To this end, we propose the Dense Reasoner (DR), which learns to perform LLM-style query reasoning directly in the embedding space at negligible cost, resulting in extremely fast retrieval.

Nevertheless, certain queries may not be adequately addressed by the structured transformation of dense reasoning. In such cases, retaining LLM-based reasoning is essential to ensure robust overall retrieval performance, thus a routing mechanism is needed to ensure appropriate reasoning process (Bai et al., 2024; Zhang et al., 2025a). To this end, we introduce the Reasoner Router (RR). Given a new query, the Reasoner Router determines whether it can be reliably handled by the Dense Reasoner, i.e., whether the learned structured transformation can stably approximate the semantic effect of LLM reasoning, and otherwise resorts to LLM for deep reasoning. This mechanism enables a controllable trade-off between efficiency and accuracy. Built on the components described above, we propose the Adaptive Query Reasoning (AdaQR) framework. AdaQR is a hybrid reasoning pipeline consisting of three components: an LLM Reasoner, a Dense Reasoner, and a Reasoner Router. The Reasoner Router flexibly directs each user query to either the Dense Reasoner or the LLM Reasoner, preserving—and in some cases improving—the retrieval quality achieved by full LLM rewrites, while substantially reducing the reasoning (see Figure 1, right).

In summary, our contributions are as follows:

- We propose a novel **AdaQR** framework that enables a hybrid fast-and-deep query reasoning strategy, preserving retrieval performance while significantly reducing rewriting cost.
- Within AdaQR, we introduce the **Dense Reasoner**, which imitates LLM reasoning in the embedding space, achieving extremely fast query rewriting.
- We further propose the **Reasoner Router** to schedule query reasoning within AdaQR. It appropriately directs each query to fast dense reasoning or deep LLM reasoning.
- Experiments on BRIGHT (Su et al., 2025) benchmark demonstrate that AdaQR generalizes well across diverse settings. Compared to serving a full LLM, our approach achieves an average 7% improvement in retrieval performance while reducing the rewriting cost by an average of 28%.

## 2 METHODOLOGY

### 2.1 PROBLEM STATEMENT

Given a query $q$, a fixed corpus $\mathcal{D} = \{d_1, d_2, \cdots, d_{|\mathcal{D}|}\}$ with $|\mathcal{D}|$ documents and a set of relevant documents $\mathcal{G}_q \subset \mathcal{D}$ of query $q$, a retrieval system $\mathcal{R}$ generates the relevance score $s_i$ for each document $d_i$ and ranks the top-k documents based on their scores, expecting that the relevant document in $\mathcal{G}_q$ appears higher in the rank list:

$$\mathcal{R}(q, \mathcal{D}) = \{(d_{i_1}, s_{i_1}), (d_{i_2}, s_{i_2}), \cdots, (d_{i_k}, s_{i_k})\}, s_{i_1} > s_{i_2} > \cdots > s_{i_k} \tag{1}$$

In modern dense retrieval, the crucial component is a powerful embedding model $\mathcal{E} : \text{text} \to \mathbb{R}^n$, mapping queries and documents into a $n$-dimensional shared continuous vector space. Let $e_q =$

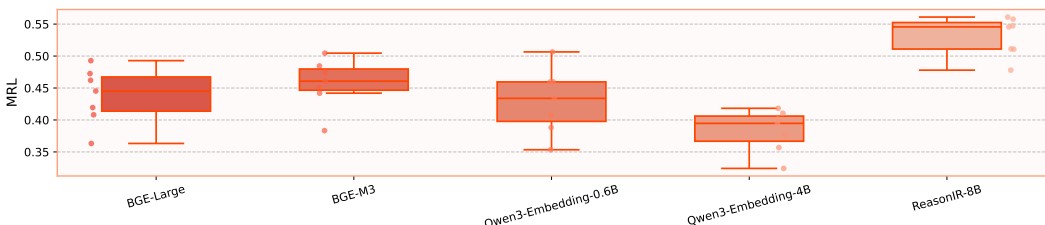

Figure 2: Distributions of MRL scores for LLM Reasoners, with each point representing the score computed for a given LLM Reasoner and dense retriever pair.

$\mathcal{E}(q)$ and $e_d = \mathcal{E}(d)$ denote the representations corresponding to query $q$ and document $d$. A dense retrieval system is then performed by computing the relevance score between $e_q$ and $e_d$ using a similarity function (e.g., cosine similarity, euclidean distance).

When meeting challenging and reasoning-intensive queries, one typically needs to rewrite them to reason their intrinsic solutions by a query reasoner, typically a powerful LLM $\mathcal{M}_{\mathrm{LLM}}$. It transforms the original query into a reasoned query in text form $q^{\mathrm{LLM}} = \mathcal{M}_{\mathrm{LLM}}(q)$. The reasoned queries typically convey richer and more explicit semantic content than the original query, and therefore replace the original query in the following retrieval steps to achieve better retrieval results.

## 2.2 PILOT STUDY

AdaQR is based on a simple, empirically informed hypothesis: for part of queries, the semantic transformation induced by LLM reasoning manifest as systematic, structured transformation in the embedding space rather than random, disordered variations.

To further verify this hypothesis, we conduct a simple experiment on BRIGHT (Su et al., 2025). Specifically, we generate the reasoned queries by employing 7 different LLM Reasoners. We then encode the original queries and all their reasoned counterparts with 5 dense retrievers. To quantify the directional coherence between the embeddings of original and reasoned queries, we compute the Mean Resultant Length (MRL) (Kutil, 2012). Concretely, for all embedding pairs $\mathcal{P} = \{(e_{q_1}, e_{q_1^{\mathrm{LLM}}}), (e_{q_2}, e_{q_2^{\mathrm{LLM}}}), \cdots, (e_{q_{|\mathcal{P}|}}, e_{q_{|\mathcal{P}|}^{\mathrm{LLM}}})\}$, MRL is defined as:

$$\mathrm{MRL}(\mathcal{P}) = \frac{1}{|\mathcal{P}|} \cdot || \sum_i \frac{e_{q_i^{\mathrm{LLM}}} - e_{q_i}}{||e_{q_i^{\mathrm{LLM}}} - e_{q_i}||} ||, \tag{2}$$

where $|| \cdot ||$ represent the L2 norm. The MRL results are summarized in Figure 2. Apparently, across all LLM Reasoners and dense retrievers, the MRL remains relatively high with an average value 0.45, indicating a substantial degree of agreement in the transformation induced by LLM-driven reasoning. Notably, different Dense Retriever exhibits different MRL value, with higher MRL value easier to learn embedding transformation, which is further corroborated in Section 3.2.

## 2.3 ADAPTIVE QUERY REASONING

Based on the above hypothesis, we propose **Ada**ptive **Q**uery **R**easoning (**AdaQR**) framework, a hybrid pipeline designed to retain the retrieval performance with LLM reasoning while dramatically reducing the reasoning cost. As shown in Figure 3, our approach presents a comprehensive framework for query reasoning, encompassing three complementary components: a LLM Reasoner, a Dense Reasoner, and an Reasoner Router.

Specifically, the LLM Reasoner is for typical query rewriting based on the LLM reasoning ability, effective but at a high cost. The Dense Reasoner is an embedding transformation optimized to approximate the semantic transformation of the LLM Reasoner's rewrites with a negligible cost, serving as an efficient alternative for LLM reasoning. Then, the reasoner router acts as a decision mechanism to route each query either to the low-cost dense reasoning or to the LLM reasoning, depending on the predictability of its LLM reasoned counterpart.

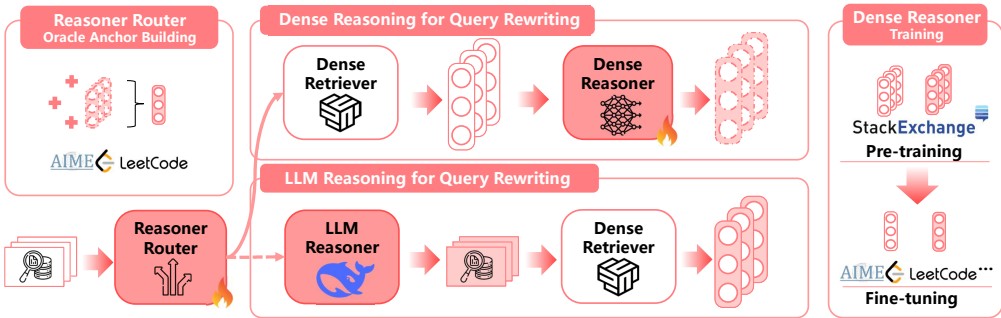

Figure 3: Overview of AdaQR: the Reasoner Router directs each input query through the oracle anchor to either the LLM Reasoner or the Dense Reasoner for reasoning-based query rewriting. The Dense Reasoner is constructed via adjacent pre-training and fine-tuning, enabling it to efficiently imitate the reasoning behavior of the LLM in the embedding space.

### 2.3.1 DENSE REASONER

The Dense Reasoner expects to reproduce the semantic transformation induced by LLM reasoning at negligible inference cost, so that its resultant embeddings can be used directly for the following retrieval steps. We achieve this by applying a compact parametric mapper and employing a two-stage training strategy.

For original query $q$ and reasoned query $q^{\text{LLM}}$ generated by LLM, along with their $d$-dimensional embedding vector $e_q$ and $e_{q^{\text{LLM}}}$, the Dense Reasoner is formalized as a parameterized mapping $\mathcal{M}_{\text{DR}}(e; \theta) : \mathbb{R}^d \to \mathbb{R}^d$ (where $\theta$ are the trainable parameters) that generates a reasoned embedding $\hat{e}_q = M_{\text{DR}}(e_q; \theta)$ intended to approximate $e_{q^{\text{LLM}}}$. The training objective is to find parameters $\theta$ minimizing the mean squared error (MSE) between predicted and target embeddings:

$$\hat{\theta} = \arg\min_{\theta} \frac{1}{M} \sum_{i=1}^{M} \left\| \mathcal{M}_{\text{DR}}(e_{v_i}; \theta) - e_{v_i^{\text{LLM}}} \right\|_2^2 \tag{3}$$

To balance generalization and domain adaptation, we adopt a **two-stage training strategy**. For the first stage, we pre-train the Dense Reasoner on large-scale embedding pairs of original and reasoned queries to learn general reasoning transformation patterns. For the second stage, to adapt to in-domain distributions while avoiding catastrophic forgetting, we fine-tune the Dense Reasoner on the downstream datasets using a reduced learning rate and epoch.

Throughout a two-stage training strategy, Dense Reasoner learns the dominant embedding transformation induced by high-quality LLM Reasoners. Benefit from the compact mapper structure, Dense Reasoner generates reasoning embeddings $\hat{v}_q$ with low-cost forward pass, avoiding LLM invocations and dense retrievers' cost while largely preserving ranking quality for queries.

### 2.3.2 REASONER ROUTER

The Dense Reasoner provides a low-cost approximation of LLM-based query reasoning, while LLM Reasoner provides higher-quality but substantially more expensive textual rewrites. The Reasoner Router is a lightweight routing mechanism that selects the most appropriate reasoning pathway for the rewriting of each query. Concretely, Reasoner Router makes a judgment whether to apply the low-cost reasoning produced by Dense Reasoner or to roll back to the LLM reasoning, enabling a controllable trade-off between total cost and retrieval effectiveness.

For each query, we use an oracle anchor to measure the predictability of its LLM reasoning result. Concretely, we consider training queries whose reasoning performance by Dense Reasoner is comparable to or better than that of LLM reasoner as $\mathcal{S}$. We build the oracle anchor for this set, which captures the shared semantic and intent signals of queries that exhibit predictable, learnable embedding-space reasoning:

$$p = \frac{1}{|S|} \sum_{q \in S} e_q \tag{4}$$

At reasoning time, the Dense Reasoner compares a query similarity $\text{sim}(e_q, p)$ to a threshold $\tau$, representing how query is likely to benefit from the low-cost Dense Reasoner. The final embedding $\tilde{e}_q$ for retrieval is defined as:

$$\tilde{e}_q = \begin{cases} \mathcal{M}_{\text{ERR}}(e_q; \hat{\theta}), & \text{if } \text{sim}(e_q, p) \geq \tau, \\ e_{q^{\text{LLM}}}, & \text{otherwise.} \end{cases} \quad (5)$$

Benefit from this flexible selection mechanism, the Reasoner Router directs each query to the most appropriate reasoning path. This component not only avoids the suboptimal retrieval caused by purely applying Dense Reasoning to unstructured queries, but also balances computational cost and retrieval quality through the adaptive threshold $\tau$ to meet resource constraints.

## 3 EXPERIMENTS

### 3.1 EXPERIMENT SETTING

**Evaluation Dataset** We employ BRIGHT (Su et al., 2025), a reasoning-intensive retrieval benchmark. It contains 1,385 real-world queries from a variety of domains (StackExchange, LeetCode, and math competitions, etc.), typically requiring deliberate semantic reasoning to match. These properties transformation BRIGHT a challenging and realistic choice for rewritten-query retrieval.

**Metrics** Following the original BRIGHT setup, we adopt the average nDCG@10 across the 12 datasets in BRIGHT as our evaluation metric. Formally, for a single query, the normalized discounted cumulative gain (nDCG) at rank $k$ is defined as:

$$\text{nDCG@}k = \frac{\text{DCG@}k}{\text{IDCG@}k} \quad \text{with} \quad \text{DCG@}k = \sum_{i=1}^{k} \frac{2^{\text{rel}_i} - 1}{\log_2(i+1)} \quad (6)$$

where $\text{rel}_i$ is the relevance of the document at rank $i$, and IDCG@k is the maximum possible DCG@k for the query.

**Pre-training of Dense Reasoner** We construct an external corpus from StackExchange [2], a popular community-driven platform. The corpus contains 10k reasoning-intensive questions (see details in Appendix A.2). We perform careful curation and filtering to ensure high-quality data and no overlap with the BRIGHT dataset. This external corpus is intended to teach the Dense Reasoner general, cross-domain reasoning transformation patterns before in-domain adaptation.

**Fine-tuning of Dense Reasoner & Oracle Anchor Building** We partition BRIGHT into an in-domain training portion and a held-out test portion, following common practice Chen et al. (2024b); Zhuang et al. (2025); Zhang et al. (2025d): 70% for fine-tune the Dense Reasoner and building oracle anchor in Reasoner Router, and 30% for evaluation.

**LLM Reasoners** We employ 17 widely-used LLMs for query reasoning as follows:

- **DeepSeek**: DeepSeek-R1 (DeepSeek-AI et al., 2025), DeepSeek-V3 (DeepSeek-AI et al., 2024), DeepSeek-R1-Distill-Qwen-32B / 14B / 7B (DeepSeek-AI et al., 2025), DeepSeek-R1-Distill-Llama-70B / 8B (DeepSeek-AI et al., 2025).
- **Qwen**: Qwen3-32B / 14B / 7B / 4B (Yang et al., 2025).
- **Meta**: LLama3-70B (Dubey et al., 2024).
- **Mistral**: Mixtral-8x7B (Jiang et al., 2024), Mistral-7B (Jiang et al., 2023).
- **ZAI**: GLM-Z1 (Zeng et al., 2024), GLM-4 (Zeng et al., 2024).
- **MoonShot**: Kimi-K2 (Bai et al., 2025).

**Dense Retrievers** We employ 5 leading dense retrieval models for encoding query and retrieval, including BGE-Large (Xiao et al., 2023), BGE-M3 (Chen et al., 2024a), Qwen3-Embedding-0.6B, Qwen3-Embedding-4B (Zhang et al., 2025c) and ReasonIR-8B (Shao et al., 2025), a SOTA retriever specialized for reasoning-intensive retrieval.

---

[2]https://huggingface.co/datasets/HuggingFaceH4/stack-exchange-preferences

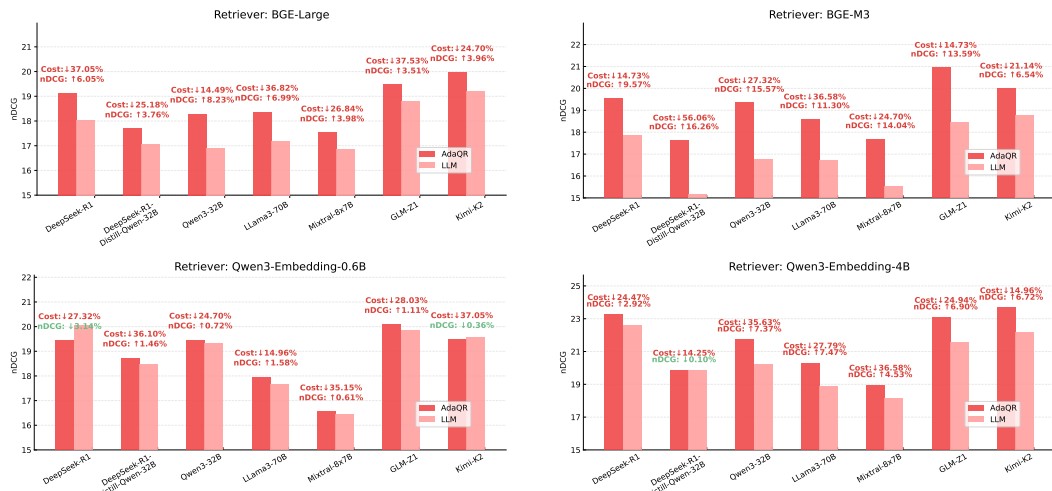

Figure 4: Retrieval performance improvement and reasoning cost reduction on the BRIGHT held-out test set achieved by AdaQR compared to LLM-based reasoning.

**Implementation Details** We implement the Dense Reasoner with a lightweight two-layer MLP. The hidden size is set equal to the input embedding dimension, and each hidden layer is followed by a tanh activation. We pre-train it for 50 epochs with a learning rate of $5 \times 10^{-4}$ and fine-tune for 3 epochs with a smaller learning rate $1 \times 10^{-5}$ to adapt to in-domain distributions. Since different dense retrievers exhibit varying representational capacities and geometric properties, the choice of the threshold parameter $\tau$ in reasoner router is different. In practice, we determine $\tau$ empirically (e.g. 0.75 for BGE-Large, 0.7 for BGE-M3 and ReasonIR-8B, 0.6 for Qwen3-Embedding-0.6B and Qwen3-Embedding-4B).

## 3.2 MAIN RESULT

We present the comparisons of AdaQR and LLM reasoning in terms of performance and cost in Figure 1 and Figure 4. Our analysis reveals several significant findings:

**AdaQR achieves better performance at lower cost.** Compared to LLM reasoning, AdaQR achieved higher nDCG in nearly all configurations, demonstrating superior retrieval performance. Take the results on ReasonIR-8B for instance, the nDCG improvement reached 22.11% for DeepSeek-R1 and 18.86% for Qwen3-32B. Concurrently, the method substantially reduced computational costs, with savings typically ranging from 15% to 37%. Across 5 dense retrievers and 7 LLM Reasoners, AdaQR achieves an average performance improvement of 7.24% while reducing costs by an average of 28.12%. This demonstrates that AdaQR achieves superior retrieval performance at a lower computational cost.

**Different dense retrievers have a significant impact on AdaQR.** For ReasonIR-8B, AdaQR delivers the most outstanding and stable performance improvement, achieving an average cost reduction of approximately 29.21% while improving nDCG performance by about 13.18%. In contrast, Qwen3-Embedding-0.6B and Qwen3-Embedding-4B show poor performance. When using these models, AdaQR delivers very limited performance gains and even exhibits slight performance decrements in some few scenarios. This phenomenon aligns with the MRL results in Section 2.2. Embedding transformation for dense retrievers with high MRL values, such as ReasonIR-8B, can be more effectively learned, thus significantly enhancing AdaQR's performance.

**AdaQR demonstrates strong generalization across LLM Reasoners and dense retrievers** AdaQR offers positive effects across 5 dense retrievers and 7 LLM Reasoners, proving itself as a broadly applicable method for LLM reasoning in text retrieval.

The results presented here are from 7 representative LLM Reasoners, while the results of remaning LLM Reasoners are reported in Appendix A.4.

| Rewriting Methods | StackExchange | | | | | | | Coding | | | Theorem-based | |
|---|---|---|---|---|---|---|---|---|---|---|---|---|
| | Bio. | Earth. | Econ. | Psy. | Rob. | Stack. | Sus. | Leet. | Pony | AoPS | TheoQ. | TheoT. |
| *Dense Retriever*: **BGE-Large** *LLM Reasoner*: **DeepSeek-R1** | | | | | | | | | | | | |
| LLM Reasoner | **46.0** | **39.0** | **24.9** | 16.8 | 8.7 | **12.6** | **18.5** | 18.5 | **0.6** | 0.3 | **18.7** | **9.9** |
| Dense Reasoner | 20.1 | 29.6 | 18.6 | 6.1 | 4.6 | 11.0 | 13.8 | **28.3** | 0.2 | **8.9** | 9.3 | 1.0 |
| *Dense Retriever*: **Qwen3-Embedding-4B** *LLM Reasoner*: **Mixtral-8x7B** | | | | | | | | | | | | |
| LLM Reasoner | **37.5** | **18.1** | **21.0** | **20.4** | **14.7** | **12.6** | **16.8** | 25.5 | **2.0** | 2.0 | 22.3 | **25.7** |
| Dense Reasoner | 28.1 | 10.6 | 14.7 | 14.6 | 6.8 | 11.3 | 12.0 | **46.1** | 0.8 | **5.8** | **39.1** | 24.5 |
| *Dense Retriever*: **ReasonIR-8B** *LLM Reasoner*: **GLM-Z1** | | | | | | | | | | | | |
| LLM Reasoner | **58.2** | **32.8** | **26.5** | **26.6** | **22.7** | **24.7** | **20.6** | 25.2 | 8.3 | 3.9 | **38.5** | **36.4** |
| Dense Reasoner | 48.0 | 28.9 | 19.0 | 21.1 | 18.2 | 17.1 | 14.1 | **38.1** | **10.4** | **7.6** | 23.6 | 23.7 |

Table 1: Performance across 3 domains with 12 tasks: Biology (Bio.), Earth Science (Earth.), Economics (Econ.), Psychology (Psy.), Robotics (Rob.), Stack Overflow (Stack.), Sustainable Living (Sus.), LeetCode (Leet.), Pony, AoPS, TheoremQA with question retrieval (TheoQ.) and with theorem retrieval (TheoT.).

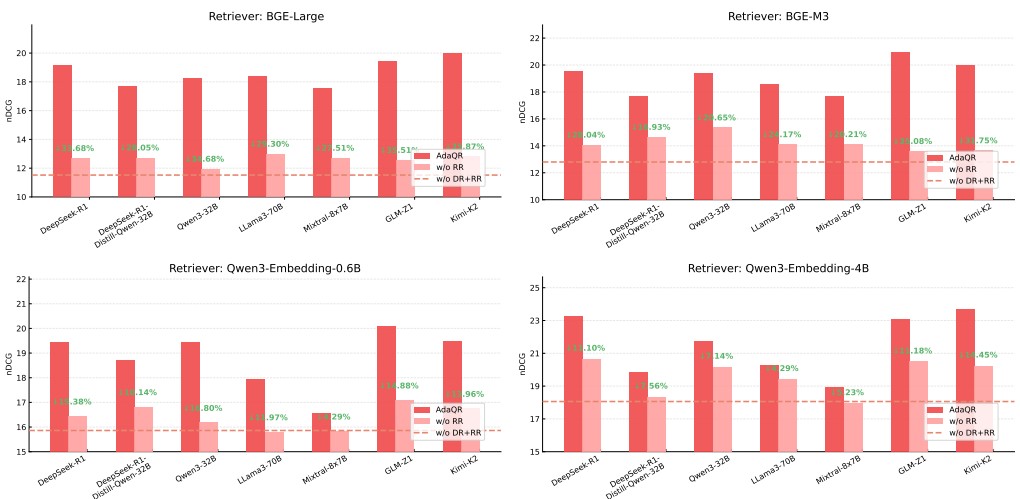

Figure 5: Ablation results after removing the Dense Reasoner and Reasoner Router components.

## 3.3 DOMAIN AND TASK SPECIALIZATIONS OF REASONERS

To further explore the performance of LLM Reasoner and Dense Reasoner on different domains and tasks, we conduct a systematic analysis across the BRIGHT benchmark, including 3 domains (StackExchange, coding and theorem-based) with 12 tasks. To ensure generality, we evaluate performance across several combinations of dense retrievers and LLM Reasoners. The result, presented in Table 1, reveals the difference across datasets.

**LLM Reasoners perform better in the StackExchange domain**. Compared with Dense Reasoners, LLM Reasoner generally show the best retrieval quality on the StackExchange domain. It demonstrates that for queries which often possesse distinctive linguistic styles and conversational conventions, LLM Reasoner can capture important surface-level or pragmatic cues, whereas Dense Reasoner struggles to fully learn these cues.

**Dense Reasoner excels in coding and theorem-based domains**. For LeetCode dataset, Dense Reasoner achieves an average nDCG of 37.5%, while LLM Reasoner only achieves 23.06%. Dense Reasoner rewriting based on embedding transformation appears to benefit from this highly structured, formalized language and task-specific terminology, which contributes most of AdaQR's outstanding performance on the full benchmark.

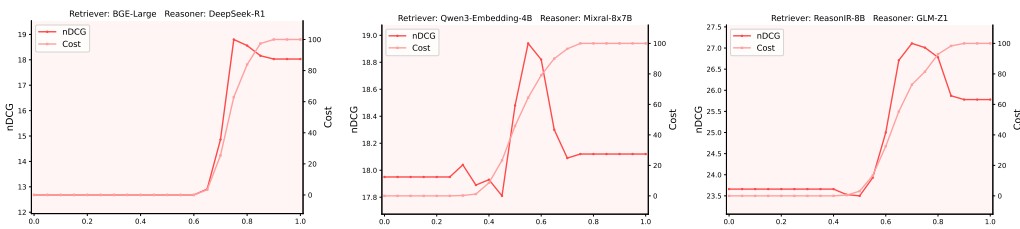

Figure 6: Performance variation of AdaQR under different values of $\tau$.

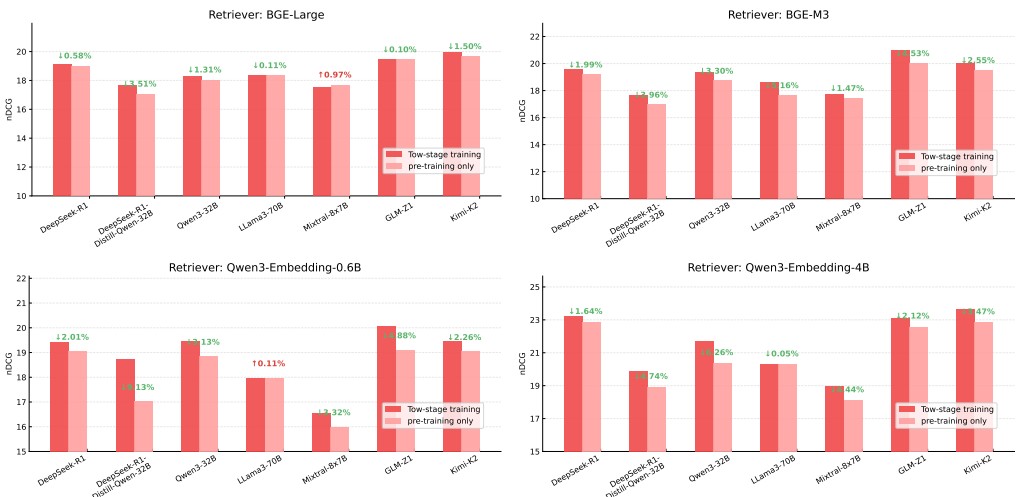

Figure 7: Performance of AdaQR under different training strategies.

## 3.4 ABLATION STUDY

To explore the individual contributions of the Dense Reasoner and the Reasoner Router, two core components within our proposed AdaQR framework, we conduct a comprehensive ablation study. Specifically, we remove Reasoner Router component, and further removed both Dense Reasoner and Reasoner Router (directly using the original query without any reasoning). Figure 5 reports the results of our ablation study: the average nDCG of AdaQR, w/o RR and w/o DR+RR are 21.05, 17.37 and 15.59 respectively, which clearly reveal the necessity of both components.

## 3.5 EFFECT OF DECISION THRESHOLD $\tau$

We investigate the effect of parameter $\tau$ (varying from 0 to 1 with a step size of 0.05), comparing performance and cost in Figure 6. When $\tau = 0$ corresponds to using the Dense Reasoner exclusively, and $\tau = 1$ corresponds to using the LLM Reasoner exclusively. The results indicate that **as $\tau$ increases, the cost increases accordingly, while nDCG exhibits a trend of increasing initially and then decreasing**. Concretely, as $\tau$ increasing, the Reasoner Router routes more queries to the LLM Reasoner, performance initially improves due to the reasoning capacity of LLMs. However, beyond a threshold (typically around 0.6 to 0.8), Reasoner Router tends to route structured queries that are better predictable by the Dense Reasoner to the LLM, resulting in higher cost and decreasing performance. This observation corroborates the view that $\tau$ and oracle anchor serves as a measure of the querys' predictability. AdaQR achieves an adaptive trade-off between performance and cost by selecting $\tau$, balancing effectiveness and efficiency.

**Reasoner Router plays a crucial role in AdaQR.** Removing Reasoner Router alone leads to a substantial performance drop, suggesting that relying solely on Dense Reasoners cannot achieve both efficiency and effectiveness. When both Reasoner Router and Dense Reasoner are removed, the performance degrades even further, confirming their complementary roles in AdaQR. The Reasoner

Router effectively predicts whether a query is predictable for DR, fully leveraging its low-cost advantage while improving retrieval performance. This dynamic allocation mechanism significantly improves both retrieval quality and efficiency.

It is worth noting that the strategy represented by w/o RR offers **a near-zero-cost query reasoning method** while still outperforming the original query approach, offering a promising option in scenarios with extremely constrained computational resources.

### 3.6 IMPACT OF TRAINING STRATEGIES

Figure 7 reports the result of the Dense Reasoner under different training strategies, comparing the proposed two-stage strategy with one that omits the fine-tuning stage. Removing the fine-tuning step leads to a slight decline in retrieval performance, with an average decrease of 2.71%. It demonstrates that the fine-tuning phase contributes to dense retrievers adapting for domain distributions, enabling more effective learning of embedding transformation. The slight decline also indicates that the Dense Reasoner has learned effective embedding transformation during the pretraining phase. Our two-stage training strategy enables the Dense Reasoner to first learn the general embedding transformation and then refine that transformation to domain-specific patterns while avoiding forgetting, thereby generating high-quality reasoning embeddings at negligible cost.

## 4 RELATED WORK

### 4.1 EFFICIENT RETRIEVAL

Driven by the need to search large corpora with low latency and high throughput, efficient retrieval has been a fundamental and persistent challenge. Efficient text retrieval has been studied in many domains, including using a novel retrieval head (Zhang et al., 2025b), learning the sequential relation between sentences to generate isomorphic embeddings (Zhang et al., 2023), creating a router to assign queries to different expert models (Lee et al., 2025), achieving end-to-end information retrieval with a single LLM (Tang et al., 2024), decomposing the input query into sub-queries to parallelize retrieval process (Zhao et al., 2025), completely removing the deep modeling of queries to maximize retrieval speed (Ma et al., 2025), and avoiding or reducing backfilling to alleviate the performance impact when switching to a new model (Ramanujan et al., 2022; Jaeckle et al., 2023).

### 4.2 QUERY REWRITING WITH LLMS

Query rewriting serves as a crucial preprocessing step, transforming short or ambiguous input queries into well-formed and optimized queries. Recent methods often leverage LLMs to enhance query understanding and expansion. For instance, Hyde, LLM4CS, and query2doc generate pseudo-documents or pseudo-responses to enrich the query context (Gao et al., 2023; Mao et al., 2023; Wang et al., 2023). QA-Expand further explores multi-agent collaboration (Seo & Lee, 2025), while InteR implements iterative information refinement between retrieval models and LLMs (Feng et al., 2024). To improve conversational search, CHIQ proposes five distinct LLM-based rewriting strategies and integrates them (Mo et al., 2024). With the continuous advancement of LLM reasoning capabilities, recently proposed Large Reasoning Models have emerged as state-of-the-art approaches for query rewriting (DeepSeek-AI et al., 2025; Yang et al., 2025; Qin et al., 2025).

## 5 CONCLUSION AND FUTURE WORK

We presented AdaQR, a hybrid query reasoning framework that combines the Dense Reasoner and Reasoner Router to achieve a balance between efficiency and retrieval quality. The Dense Reasoner efficiently approximates LLM reasoning in the embedding space, while the Reasoner Router adaptively directs queries to ensure robustness on challenging cases. Experiments on large-scale retrieval benchmarks show that AdaQR maintains or improves retrieval performance compared to full LLM rewrites, while significantly reducing computation. Future work includes enhancing dense reasoning strategies, extending AdaQR to multi-modal retrieval, refining the routing mechanism, and evaluating its performance in real-world high-throughput systems.

## ETHICS STATEMENT

This work adheres to the ICLR Code of Ethics. Our study focuses on improving efficiency and effectiveness in information retrieval systems and does not involve human subjects or personally identifiable information. All datasets used are publicly available benchmark datasets, and we comply with their respective licenses and usage guidelines. The proposed AdaQR framework is intended to enhance retrieval performance and reduce computational cost, and we do not foresee any direct harmful societal impacts arising from its use. Potential ethical considerations include bias in retrieval outcomes due to the underlying pre-trained language models or dataset distributions. We encourage users to be aware of such biases when deploying retrieval systems in sensitive applications. We also ensure transparency in our experimental methodology, including dataset usage, model configurations, and evaluation protocols, to support reproducibility and research integrity. Finally, no conflicts of interest or external sponsorships influenced the work reported in this paper.

## REPRODUCIBILITY STATEMENT

We have made every effort to ensure the reproducibility of our results. All datasets used in our experiments are publicly available benchmark datasets, and detailed descriptions of dataset processing, splits, and evaluation metrics are provided in the main paper and supplementary materials. The implementation details of the AdaQR framework, including the Dense Reasoner and Reasoner Router components, as well as hyperparameters, training procedures, and model architectures, are documented in the article.

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

## A  APPENDIX

### A.1  LLM USAGE

In this work, we used ChatGPT (GPT-5) as an assistive tool for drafting and refining text in the introduction and related work. All content produced with the assistance of ChatGPT was reviewed, revised, and verified by the authors. ChatGPT contributed to wording suggestions and phrasing improvements but did not contribute independently to research ideation, experimental design, or result analysis. The authors take full responsibility for all content in this paper.

### A.2  DATASETS

For StackExchange for Dense Reasoner pre-training, we collect 9795 queries from 17 domains: ai, biology, bioinformatics, chemistry, codereview, computergraphics, cs, earthscience, economics, math, mathoverflow, philosophy, physics, robotics, stackoverflow, softwareengineering, sustainability. Each dataset contributes 600 queries, except for computergraphics (364), philosophy (599) and sustainability (432). During the collection phase, we excluded queries containing images and selected only those whose answers were chosen. We also carefully reviewed all candidate queries to ensure no overlap with queries from the BRIGHT benchmark. The prompt we use for query reasoning is shown in Figure 10.

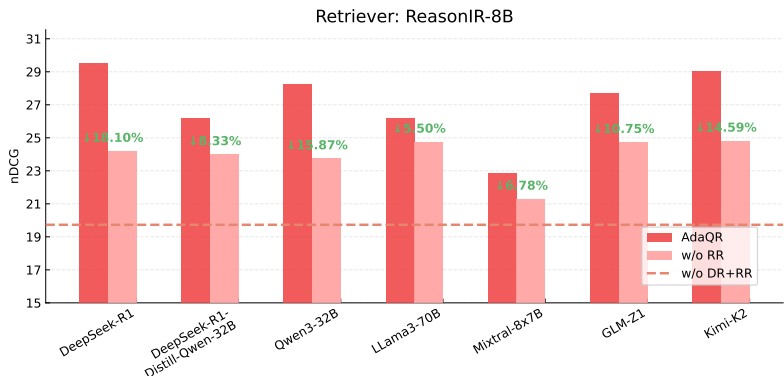

Figure 8: Ablation results on components of ReasonIR-8B

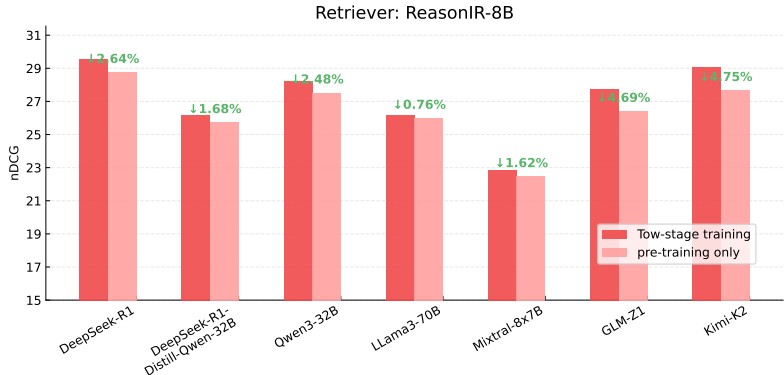

Figure 9: Performance of AdaQR under different training strategies of ReasonIR-8B

### A.3 RESULTS OF REASONIR-8B

We present results of ReasonIR-8B in Figure 8 and Frigure 9, which are not presented in Section 3.

### A.4 MAIN RESULTS OF ADDITIONAL LLMS

Fig 11 reports the retrieval performance improvement and reasoning cost reduction achieved by AdaQR compared to LLM-based reasoning across 10 additional LLMs.

> **Prompt template for LLM reasoning**
>
> **{Query}**
>
> **Instructions:**
>
> **1.  Identify the essential problem.**
>
> **2.  Think step by step to reason and describe what information could be relevant and helpful to address the questions in detail.**
>
> **3.  Draft an answer with as many thoughts as you have.**

Figure 10: Prompt template used to guide the LLM Reasoner to thoroughly analyze and provide detailed explanations for a given query.

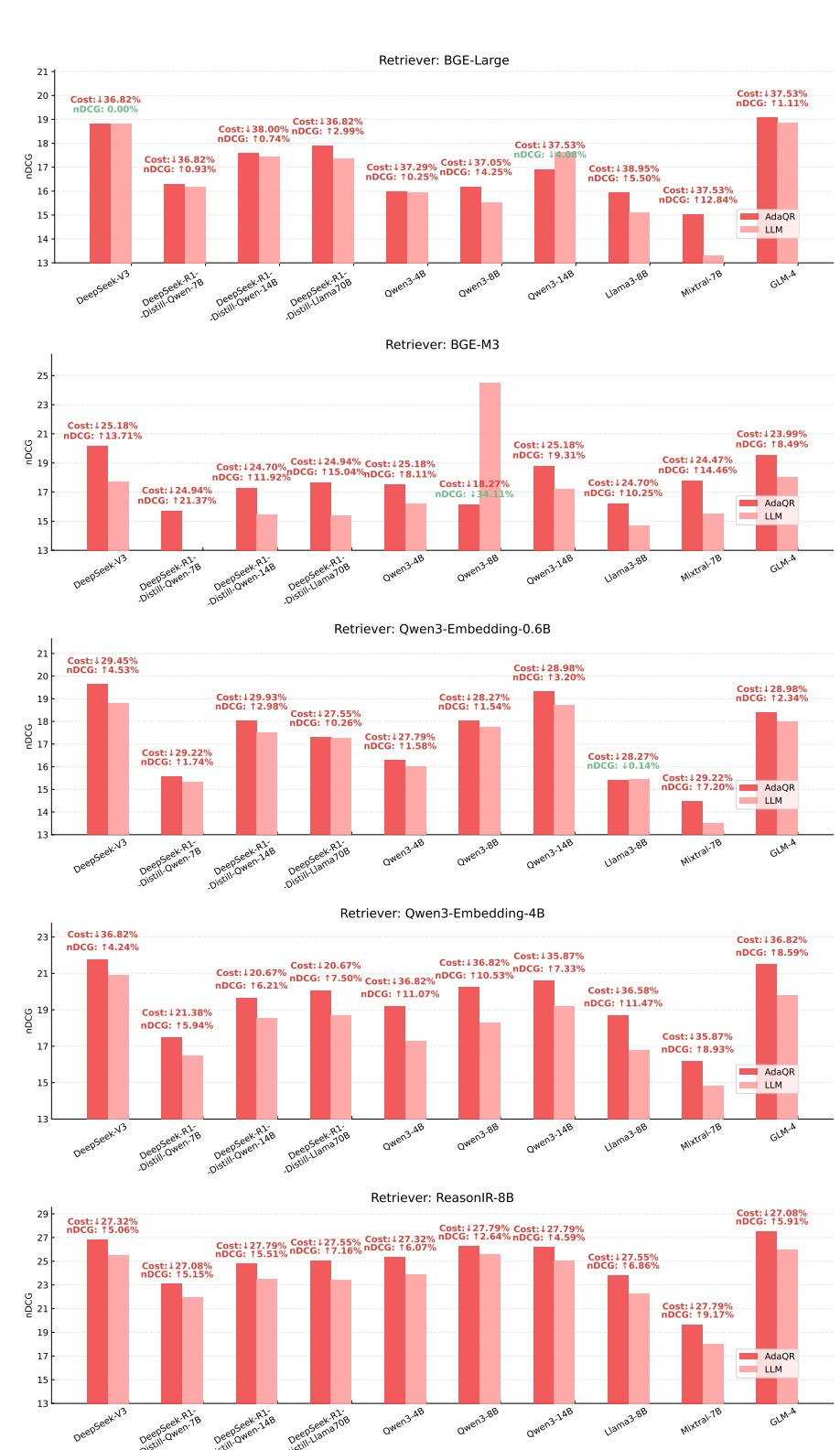

Figure 11: Retrieval performance improvement and reasoning cost across 10 additional LLMs

