# OpenReview forum: "Your Dense Retriever is Secretly an Expeditious Reasoner"
_ICLR.cc/2026/Conference — ICLR 2026 Conference Withdrawn Submission_

### Official Review · Reviewer_tFrr · 2025-10-27

**Soundness:** 2
**Presentation:** 2
**Contribution:** 3
**Rating:** 2
**Confidence:** 4

**Summary:**

This paper presents AdaQR, a framework to improve the efficiency of query rewriting. The main idea is that for in-distribution queries, the system can directly map a query embedding to its corresponding “reasoned query” embedding. Normally, this would require using an LLM to generate a reasoning chain and then embedding that chain with the embedding model. Experiments on BRIGHT demonstrate both performance and cost benefits for the proposed approach.

**Strengths:**

1. The idea is novel. Directly navigating the query embedding space to obtain a reasoned query embedding is interesting and new. The paper provides solid empirical results demonstrating advantages in both efficiency and performance.
2. The motivation is strong. The pilot study analyzes the mean resultant length between the original query embedding and the transformed query embedding across seven different reasoners and five different embedding models, and shows strong alignment. This offers empirical evidence supporting the core idea of the paper.

**Weaknesses:**

1. Compared to an out-of-the-box retrieval framework, the method introduces new modules that require training for each LLM reasoner/embedding model combination.
2. The method adds computational overhead. Each retrieval now includes: embedding the query for the Reasoner Router to decide between LLM-enabled query rewriting and the Dense Reasoner; if routed to the dense path, (1) embed the query and (2) apply the Dense Reasoner to the query; if routed to the LLM path, (1) generate a reasoning chain with the LLM and (2) embed the enhanced query.
3. The paper does not clearly define how computational cost is measured. Although cost reduction is mentioned in several places, it is unclear whether this refers to end-to-end latency, FLOPs, dollar cost, or another metric. It is also unclear how this cost varies with the choice of LLM reasoner and embedding model.
4. The BRIGHT data split is unclear. The baseline LLM reasoning approach has no trainable components, while AdaQR does. It is therefore unsurprising that training on BRIGHT could improve nDCG relative to out-of-the-box LLM reasoners. I am concerned that the benefit may come from overfitting to BRIGHT’s query styles. BRIGHT is a meta-dataset with multiple data sources; lines 250–253 do not make it clear whether the 70/30 split is stratified by source or simply a random split across all instances.

**Questions:**

- The pilot study is helpful, but how should we conceptually understand the phenomenon of performing reasoning directly in the embedding space? An analysis of the router’s decisions—i.e., what types of queries are sent to the Dense Reasoner versus the LLM reasoner—would help interpret what the Dense Reasoner has learned.
- Conceptually, the Dense Reasoner seems to standardize queries and map them into a more canonical direction. It is unclear how much genuine “reasoning” it performs beyond normalization or transformation of embeddings.
- What is the architecture of the Dense Reasoner? I could not find a detailed description of its implementation in the paper.
- I was not able to find implementation and training details for the Dense Reasoner and the router in the appendix. Please provide comprehensive details (model architectures, training objectives, hyperparameters, and compute budget) to enable reproducibility.

---

> ### Author Response · Authors · 2025-11-20
> **Response4.1: generalization of AdaQR, how computational cost is measured, cost variation and increased computational cost**
>
> We are deeply grateful for your thorough review of our work. We clarify the following points:
>
> **1. Generalization of AdaQR and the BRIGHT 70/30 split**
>
> * The BRIGHT 70/30 split
>
> This split is stratified by **source** (Biology, Leetcode, TheoremQuestion……), which is omitted this detail in the original manuscript. Thank you for pointing it out.
>
> * Overfitting and generalization of in-domain fine-tuning
>
> Reviewer **SByg** raised the same point. We address it in detail in the **Response2.2**: generalization of AdaQR and other reasoning-intensive retrieval models (1. Generalization of AdaQR).
>
>
>
>
> **2.  Computation cost and cost variation**
>
> The original manuscript is indeed lack of a clear, quantitative description of computational cost. In the initial submission we reported cost changes using **simple percentages** because different LLMs incur very different absolute costs. For example, if 30% of queries are routed to the Dense Reasoner, we reported a “30% cost reduction” relative to a full LLM Reasoner. While this percentage-based view is simple and intuitive, it lacks concrete units and can be misleading when comparing across LLM providers and embedding backbones.
>
> In the updated version, we have systematically presented **dollar* cost as the computational cost because our LLM Reasoner is accessed via provider APIs. We have also analyzed the cost variation with the choice of LLM reasoner and embedding model.
>
> * **Dollar cost**
>
> As shown below, we report the dollar cost on the test set for various LLM Reasoner and Dense Retriever (take BGE-Large and ReasonIR-8B as examples). The average cost of LLM Reasoner is \$282, while AdaQR surpasses LLM by an average of **7%** in NDCG across five embedding models, at a cost of just **\$192**.
>
> | LLM Reasoner        | DeepSeek-R1 | DeepSeek-R1-Distill-Qwen-32B | Qwen3-32B | LLama3-70B | Mixtral-8x7B | GLM-Z1 | Kimi-K2 |
> |---------------------|-------------|------------------------------|-----------|------------|--------------|--------|---------|
> | LLM                 | \$482        | \$189                         | \$252      | \$228       | \$164         | \$156   | \$504    |
> | AdaQR (BGE-Large)   | \$317        | \$164                         | \$215      | \$146       | \$121         | \$102   | \$379    |
> | AdaQR (ReasonIR-8B) | \$282        | \$120                         | \$148      | \$149       | \$132          | \$98    | \$373    |
>
> * Cost variation
>
>   * LLM Reasoner: DeepSeekR1 and Kimi-K2 incur the highest dollar cost due to their higher per-token prices and longer outputs, though they also deliver the strongest performance. In contrast, GLM-Z1 and Mixtral-8x7B are the cheapest, with GLM-Z1 offering particularly strong cost-effectiveness as its performance is only slightly below the top models.
>
>   * Embedding model: The trend matches the Section 2.1 pilot study: ReasonIR-8B, which has the highest MRL, achieves the lowest cost under AdaQR. However, differences across embedding models are modest compared to LLM choices, and overall cost remains strongly influenced by the router's similarity-threshold setting.
>
> We hope these additions address your concern about quantifying compute cost.
>
> **3. Increased computational cost.**
>
> We appreciate you raising this important concern. We agree that AdaQR introduces additional computation. However, our comparisons are against **LLM-based query rewriting**, and for reasoning-intensive retrieval the benefits of rewriting are substantial. For example, a vanilla out-of-the-box retrieval run with Qwen3-Embedding-0.6B obtains nDCG 15.86, whereas AdaQR raises this to 19.82 in our experiments. More broadly, we observe that rewriting yields significant relative gains (**20%–60%**) in reasoning-intensive queries, which sufficiently justifies the additional overhead.
>
> Besides, some of the added cost is necessary: computing query embeddings is required by any dense retrieval pipeline (including the out-of-the-box baseline). The incremental cost introduced by the Dense Reasoner itself is minimal. It is a tiny MLP and its inference cost is negligible compared with LLM calls. In fact, using the Dense Reasoner alone (no LLM) yields an average nDCG improvement of roughly **5%**, showing that the Dense Reasoner by itself provides meaningful benefit at very low cost.
>
> In short, while AdaQR does increase computation relative to a bare retrieval system, the substantial retrieval-quality gains on reasoning-intensive tasks make this extra cost unavoidable but worthwhile. Moreover, compared with the same LLM-rewriting approach, AdaQR achieves both lower monetary cost and higher retrieval performance by routing many queries to the lightweight Dense Reasoner.

---

> ### Author Response · Authors · 2025-11-20
> **Response4.2: missing or unclear details of Dense Reasoner and Reasoner Router**
>
> We are deeply grateful for your meticulous and professional feedback, and we have systematically rewritten **Section 2.3.1** (Dense Reasoner) and **Section 3.1** (Experiment Setting) to ensure that every ambiguity is resolved. We will publicly release the code and datasets.
>
> For your convenience, we clarify the specific issues you raised as follows:
>
> **1. Model Architecture**
>
> Our Dense Reasoner is a compact multilayer perceptron (MLP) that maps query embeddings to reasoned-query embeddings. Concretely, the network consists of an input layer, two hidden layers, and an output layer. Both the input and the output are $d$-dimensional vectors, where $d$ is the embedding dimensionality of the chosen dense retriever (for example, $d=1024$ for BGE-M3). The shape of each hidden layer is $(d, d)$, and we apply a Tanh activation after each hidden layer.
>
> **2. Training objective**
>
> As described in Section 2.3.1, Dense Reasoner is trained to produce outputs that are close to the ground-truth embeddings of the LLM-reasoned queries. We optimize mean squared error (MSE) between the model output and the target LLM embedding, as presented in formula (3)
>
> **3. Training details**
>
> Pretraining is run for 50 epochs with a learning rate $5 \times 10^{-4}$, while in-domain fine-tuning is run for 3 epochs with a learning rate $1 \times 10^{-5}$.
>
> **4. Computing cost**
>
> Our experiments are conducted on a single NVIDIA GeForce RTX 3090 GPU. The Dense Reasoner is extremely lightweight, and the full pretraining plus in-domain fine-tuning process completes in only a few minutes on this hardware, with virtually no waiting time during inference, resulting in negligible compute cost compared with LLM-based rewriting. For all stages we used a batch size of 64, the Adam optimizer, and an early-stopping mechanism. Random seed 42 is used for all experiments to ensure reproducibility.
>
> **5. Reasoner Router**
>
> Reasoner Router first calculates an oracle anchor $p$ during in-domain fine-tuning stage, by averaging
> the training queries whose reasoning performance by Dense Reasoner is comparable to or better than that of LLM reasoner. For a query $e_q$, the cosine similarity is compared with the threshold $\tau$ to router to the most appropriate reasoning path. as presented in formula (4) and (5).

---

> ### Comment · Reviewer_tFrr · 2025-11-24
>
> Thank you for the detailed response. I have raised my score to 4.
>
> "1. Generalization of AdaQR and the BRIGHT 70/30 split": Since the LLM is trained on general corpus, wouldn't it be unfair to compare the performance of AdaQR vs LLM in Figure 4? I think splitting the data by data source would greatly strengthen the argument of the paper.
>
> "2. Computation cost and cost variation": Does the cost here only cover the inference cost, or does it also include the training cost?

---

> > ### Author Response · Authors · 2025-11-26
> > **Response to remaining concerns**
> >
> > Thank you for pointing them out. We clarify the following points:
> >
> > **1. Generalization of AdaQR and the BRIGHT 70/30 split.**
> >
> > First, we would like to emphasize that AdaQR is designed as an add-on framework rather than a complete replacement for LLMs. Whether the LLM is trained on a general corpus or domain-specific data, AdaQR uses the LLM as one component within a larger system. Our goal with AdaQR is to reduce inference cost while simultaneously improving retrieval performance. Therefore, the comparison between AdaQR and LLM is not about replacement but about enhancing LLMs' efficiency in a retrieval system. Thus, questions of "fairness" between the two are not central.
> >
> > Regarding your point about splitting the data by data source: as we mentioned earlier, our 70/30 split is stratified by source (e.g., Biology, Leetcode, TheoremQuestion, etc.), meaning that 70% of data from each source was used for training, and 30% for testing, rather than a random split of the entire BRIGHT dataset. We apologize if this was unclear in the earlier description.
> > We believe you may be suggesting an additional level of splitting, where we use all data from 70% of the sources for training and 30% for testing, which would better validate generalization across sources. We have conducted this experiment, and the results show that, even with this source-based split, the average NDCG improvement is **5%** compared to pretraining. When compared to AdaQR with domain-specific fine-tuning, the performance is slightly lower, but still demonstrates AdaQR's strong generalization ability across different sources.
> >
> > **2. Computation cost and cost variation**
> >
> > we apologize for not clearly explaining the cost structure of AdaQR in our initial submission. To clarify, the cost we refer to is specifically for inference cost, not the training cost.
> >
> > In terms of training costs, our labeled dataset for training the Dense Reasoner is approximately 5 times the cost of LLM-based rewriting. However, it is important to note that while there exists training cost, this cost is one-time and can be reused across different embedding models and queries. Therefore, the training cost can be amortized over multiple deployments, making it a more sustainable solution in the long term with net savings. In contrast, LLM-based rewriting incurs recurring costs with every query processed, making it a more expensive long-term solution.
> >
> > We hope these resolve your concerns.

---

### Official Review · Reviewer_1xNL · 2025-10-28

**Soundness:** 2
**Presentation:** 3
**Contribution:** 2
**Rating:** 6
**Confidence:** 4

**Summary:**

To address the high cost of using LLMs for query rewriting in dense retrieval, the authors propose AdaQR. This framework dynamically routes queries: dynamically send queries to either a full LLM or a lightweight "Dense Reasoner" that operates in the embedding space, balancing performance and efficiency.

**Strengths:**

1. The design of the Dense Reasoner is interesting. By directly learning to approximate the embeddings of LLM-rewritten queries, it provides a lightweight rewriting mechanism
2. The Router appears capable of balancing the trade-off between the efficient but less accurate Dense Reasoner and the high-performance but costly LLM-based method.

**Weaknesses:**

According to Figure 6, AdaQR is shown to outperform both the LLM-only and Dense Reasoner-only baselines. This is surprising, as the performance of a router-based method like AdaQR is typically bounded by its components. This result suggests that the LLM and Dense Reasoner are complementary, with the router sending queries the LLM fails on to a successful Dense Reasoner. This I believe somehow contradicts the core design, where the Dense Reasoner is trained to mimic the LLM. The paper needs a deeper analysis to resolve this paradox and explain how a model trained for imitation can correct its teacher.

**Questions:**

See weaknesses

---

> ### Author Response · Authors · 2025-11-20
> **Response to the complementarity of Dense Reasoner and LLM Reasoner**
>
> Thank you for raising this interesting and important question about why AdaQR can outperform both the LLM-only and Dense-Reasoner-only baselines. We clarify the following points:
>
> **1. The Dense Reasoner is not strictly bounded by the teacher.**
>
> Although the Dense Reasoner is trained to approximate LLM-rewritten embeddings, distillation-like training does not imply that a student model can only imitate its teacher. Prior work has shown that compact or distilled models can reorganize or expand reasoning paths and even surpass the teacher [1,2]. This suggests that the Dense Reasoner may capture complementary or enhanced reasoning patterns beyond those utilized by the LLM.
>
> **2. Embedding-level mapping filters noisy surface signals.**
>
> The Dense Reasoner operates in embedding space rather than at the token level, which naturally filters out noisy or low-quality details in LLM rewrites. This reduces overfitting to LLM-specific phrasing and can improve retrieval quality. In our experiments, a small portion of LLM rewrites were ineffective or even harmful, and on these cases the Dense Reasoner typically performs better.
>
> **3. Domain-dependent complementarity.**
> As discussed in Section 3.3 and shown in Table 1, the Dense Reasoner performs exceptionally well on coding and theorem-style queries. We believe that this advantage benefits from the structured nature of these domains, and it partially explains the complementary behavior between the two models. We plan to explore this phenomenon more systematically across additional datasets in future work.
> The Dense Reasoner and LLM complement each other, which is why AdaQR achieves better performance than either model alone. We hope these clarifications address your concern.
>
> [1] Yao Huang et al. De-
> ceptionbench: A comprehensive benchmark for ai deception behaviors in real-world scenarios.
> Advances in Neural Information Processing Systems.
>
> [2] Zhen Huang et al. O1 replication journey–part 2: Surpassing o1-preview
> through simple distillation, big progress or bitter lesson? arXiv preprint arXiv:2411.16489, 2024.

---

> > ### Comment · Reviewer_1xNL · 2025-11-22
> > **Response to Authors**
> >
> > No, my concern remains unaddressed. Could just you provide a more in-depth analysis explaining why AdaQR outperforms both the LLM-only and Dense-Reasoner-only baselines? Also, under what conditions does the Dense-Reasoner excel compared to the LLM, and conversely, in which scenarios does the LLM demonstrate superior performance?
> >
> > To be honest, the proposed routing mechanism appears to align closely with the 'learning to defer' research. While the design of the Dense-Reasoner itself is interesting,  a more rigorous analysis focused specifically on the router's behavior and decision-making process is important.

---

### Official Review · Reviewer_SByg · 2025-11-01

**Soundness:** 2
**Presentation:** 2
**Contribution:** 2
**Rating:** 6
**Confidence:** 4

**Summary:**

To improve the efficiency of reasoning-intensive retrieval, this paper proposes adaptive query reasoning (AdaQR). The framework introduces a Dense Reasoner and a Reasoner Router to the retrieval pipeline. In particular, the Dense Reasoner transforms the embedding from the original query space to a reasoned query space. This bypasses the costly LLM rewriter + dense retriever approach. As the Dense Reasoner is small, a Reasoner Router is introduced to check whether the query should go through the Dense Reasoner or the LLM Reasoner. Empirically, on the BRIGHT reasoning-intensive retrieval dataset, AdaQR achieves significant cost reduction while surprisingly also results in improved performance.

**Strengths:**

1. Training a dense reasoner to induce reasoned embedding is a very cool idea. Pairing it with a router is capable of leveraging its efficiency benefit while even improving performance.
2. It’s quite surprising to see that sometimes the dense reasoner can offer advantages over the LLM reasoner, so sometimes the queries just should not be rewritten at all, which can somewhat be captured by this dense reasoner.
3. The evaluations and ablation studies are comprehensive with promising gains.
4. The paper is generally well-written.

**Weaknesses:**

1. The Dense Reasoner requires accessing 70% BRIGHT’s ground-truth query and document as training data. The Reasoner Router requires knowing the in-domain query embedding as the Oracle Anchor beforehand. Both prevent the framework from generalizing to unseen domains.
2. Some details are missing. For example, (a) it is unclear how the cost is calculated, (b) it is unclear how to ensure the pretraining corpus has no overlap with queries from BRIGHT.  Providing more details about that would make the results more convincing.

**Questions:**

1. How much traffic is routed to dense reasoner with respect to $\tau$? It would be useful to visualize this with a plot to see more details about the efficiency gain.
2. The dense reasoner has a very small size. Do the authors have suggestions on how its size affects the performance and how its size should be determined?
3. How does the Dense Reasoner perform if it is only pretrained without in-domain adaptation on ReasonIR? Does it also have only a slight decline?
4. Other than ReasonIR-8B, has the author tried other reasoning-intensive retrieval models?

---

> ### Author Response · Authors · 2025-11-20
> **Response 2.1: computational cost, size of Dense Reasoner, data over-lap and traffic with respect to $\tau$**
>
> **1. How computational cost is measured**
>
> The original manuscript is indeed lack of a clear, quantitative description of computational cost. In the initial submission we reported cost changes using **simple percentages** because different LLMs incur very different absolute costs. For example, if 30% of queries are routed to the Dense Reasoner, we reported a "30\% cost reduction" relative to a full LLM Reasoner. While this percentage-based view is simple and intuitive, it lacks concrete units and can be misleading when comparing across LLM providers and embedding backbones.
>
> In the updated version, we have systematically presented **dollar cost** as the computational cost because our LLM Reasoner is accessed via provider APIs. As shown below, we report the dollar cost on the test set for various LLM Reasoner and Dense Retriever (take BGE-Large and ReasonIR-8B as examples). The average cost of LLM Reasoner is \$282, while AdaQR surpasses LLM by an average of **7%** in NDCG across five embedding models, at a cost of just **\$192**.
>
> We hope these additions address your concern about measuring compute cost.
>
> | LLM Reasoner        | DeepSeek-R1 | DeepSeek-R1-Distill-Qwen-32B | Qwen3-32B | LLama3-70B | Mixtral-8x7B | GLM-Z1 | Kimi-K2 |
> |---------------------|-------------|------------------------------|-----------|------------|--------------|--------|---------|
> | LLM                 | \$482        | \$189                         | \$252      | \$228       | \$164         | \$156   | \$504    |
> | AdaQR (BGE-Large)   | \$317        | \$164                         | \$215      | \$146       | \$121         | \$102   | \$379    |
> | AdaQR (ReasonIR-8B) | \$282        | \$120                         | \$148      | \$149       | \$132          | \$98    | \$373    |
>
> **2. The size of the Dense Reasoner**
>
> We conducted ablation experiments over Dense Reasoner depth (1–5 layers) and hidden-layer dimensionality ($\frac{d}{2}$、$d$、 $2d$、 $d+512$) to study how model size affects performance. The results show that hidden-layer width has only a minor effect, whereas depth matters substantially: a single-layer MLP is generally too shallow to perform the embedding-space reasoning well, while increasing depth yields diminishing returns after a point. Balancing retrieval gains against parameter count and inference cost, we therefore selected a two-layer MLP with hidden size $d$ as the final Dense Reasoner architecture, which strong performance with minimal added compute.
>
> **3. Data contamination to ensure no over-lap**
>
> Thank you for the careful comment. We omit the data-contamination check in the original appendix. Ensuring no overlap between our pretraining corpus and BRIGHT is a fundamental prerequisite for our experiments. We implement a two-stage contamination test pipeline and add a full description to the appendix.
> First, we embed both the BRIGHT queries and all pretraining documents with BGE-M3 and compute pairwise **embedding similarity**. Any pretraining document whose similarity to a BRIGHT query exceeds 0.8 is flagged as a candidate for closer inspection. Second, following the methodology used in LLaMA-style contamination checks [1], we compute **n-gram overlap** between flagged candidates and BRIGHT queries and then manually review the hits. Items judged to be high-risk by manual review have been removed from the pre-training set. This two-stage procedure makes accidental leakage extremely unlikely.
>
> **4. How much traffic is routed to dense reasoner with respect to** $\tau$
>
> Thank you for the question. We analyze routing traffic in Section 3.5 (Figure 6), where we perform a detailed ablation over the router similarity threshold $\tau$. In the figure the Cost column reports the fraction of queries routed to the LLM Reasoner (i.e., the complement of the fraction handled by the Dense Reasoner). As $\tau$ increases the router becomes more conservative: fewer queries are sent to the Dense Reasoner and more are forwarded to the LLM. The largest change occurs in the $\tau\in[0.6,0.8]$ range, covering roughly 80\% of queries. Hope this addresses your concern.
>
> [1]  Zhangir Azerbayev, Hailey Schoelkopf, Keiran Paster, Marco Dos Santos, Stephen Marcus McAleer, Albert Q. Jiang, Jia Deng, Stella Biderman, and Sean Welleck. Llemma: An open language model for mathematics. In The Twelfth International Conference on Learning Representations, ICLR 2024, Vienna, Austria, May 7-11, 2024.

---

> ### Author Response · Authors · 2025-11-20
> **Response2.2: generalization of AdaQR and other reasoning-intensive retrieval models**
>
> **1. Generalization of AdaQR**
>
> We sincerely agree this is critical. We omit the domain-specific fine-tuning stage for Dense Reasoner, relying solely on pre-training. During pre-training, we computed the router's oracle anchor instead of domain-specific anchor to validate AdaQR's generalization. Section 3.6 (Figure 7) shows detailed results of the ablation. The ablation results of **ReasonIR-8B** which is not presented in the initial manuscript is shown as follow, with an **slight decline of 2.66\%**.
>
> Compared to AdaQR with domain-specific fine-tuning, there is a slight performance decline, with the average NDCG decreasing of 2.71\%. However, compared to the LLM-rewritten NDCG 20.57, it still achieved a **4.53%** improvement. Even without in-domain fine-tuning the Dense Reasoner, AdaQR still delivers strong gains over direct LLM-based rewriting, demonstrating that the method benefits are not solely driven by domain-specific adaptation. The full AdaQR pipeline does gain additional, measurable improvement from in-domain fine-tuning on BRIGHT's query style, but this improvement is not dominant. Overall, these results indicate good out-of-domain robustness and practical generalizability of AdaQR. We hope the above clarifications and ablations address your concerns.
>
> | LLM Reasoner        | DeepSeek-R1 | DeepSeek-R1-Distill-Qwen-32B | Qwen3-32B | LLama3-70B | Mixtral-8x7B | GLM-Z1 | Kimi-K2 |
> |---------------------|-------------|------------------------------|-----------|------------|--------------|--------|---------|
> | AdaQR               | 29.55       | 26.18                        | 28.23     | 26.20      | 22.85        | 27.72  | 29.06   |
> | AdaQR w/o fine-tune | 28.77       | 25.74                        | 27.53     | 26.00      | 22.48        | 26.42  | 27.68   |
>
> **2. Performance on other reasoning-intensive retrieval models**
>
> Thank you for raising this insightful question. To further assess generalization, we have conducted additional experiments on DIVER [1]. DIVER is the 1st reasoning-intensive retriever on the BRIGHT benchmark. Specifically, we evaluated Diver-Retriever-0.6B and Diver-Retriever-4B. As shown in the table below, applying AdaQR to both models yields substantial nDCG improvements over the LLM-only rewriter (**24.53 to 25.30 and 27.25 to 28.24**), while simultaneously reducing the average inference cost (**34.27\% and 46.93\%**). These results demonstrate that AdaQR not only performs well across different retriever architectures, but also generalizes strongly to state-of-the-art reasoning-intensive retrieval models.
>
>
> | LLM Reasoner                        | DeepSeek-R1 | DeepSeek-R1-Distill-Qwen-32B | Qwen3-32B | LLama3-70B | Mixtral-8x7B | GLM-Z1 | Kimi-K2 |
> |-------------------------------------|-------------|------------------------------|-----------|------------|--------------|--------|---------|
> | LLM Reasoner (Diver-Retriever-0.6B) | 26.42       | 22.92                        | 25.12     | 23.69      | 22.63        | 24.73  | 26.22   |
> | AdaQR (Diver-Retriever-0.6B)        | **26.88**       | **23.63**                        | **25.96**     | **25.18**      | **23.27**        | **25.93**  | **26.23**   |
> | AdaQR (Diver-Retriever-0.6B) (Cost) | 34.2        | 34.44                        | 33.97     | 34.2       | 33.25        | 35.39  | 34.44   |
> | LLM Reasoner (Diver-Retriever-4B)   | 27.85       | 26.50                         | 28.01     | 26.93      | 25.65        | 27.41  | 28.41   |
> | AdaQR (Diver-Retriever-4B)          | **28.60**        | **27.25**                        | **28.69**     | **27.92**      | **27.05**        | **29.25**  | 28.91   |
> | AdaQR (Diver-Retriever-4B) (Cost)   | 47.74       | 46.56                        | 47.74     | 47.03      | 46.79        | 45.84  | 46.79   |
>
>
> [1] Meixiu Long, Duolin Sun, Dan Yang, Junjie Wang, Yecheng Luo, Yue Shen, Jian Wang, Hualei Zhou, Chunxiao Guo, Peng Wei, Jiahai Wang, Jinjie Gu. "Diver: A multi-stage approach for reasoning-intensive information retrieval." arXiv preprint arXiv:2508.07995, 2025.

---

> > ### Comment · Reviewer_SByg · 2025-11-22
> >
> > 1. Thanks for addressing most of my concerns. Could you elaborate more on the effect of the size of the Dense Reasoner with actual statistics?
> >
> > 2. Appreciate the authors adding experiments with DIVER. Also, you probably meant that DIVER is the top-performing reasoning-intensive retriever instead of the 1st. In addition, according to the leaderboard, DIVER retriever (4B) was able to achieve performance around 31.9 on the original query (w/o LLM rewriting), but the reported performance here seems to be lower. Are these numbers accurate?

---

> > > ### Author Response · Authors · 2025-11-24
> > > **Response to remaining concerns**
> > >
> > > Thank you for pointing them out. We clarify the following points:
> > >
> > > **1.Statistics of the size of the Dense Reasoners**
> > >
> > > We ran systematic ablations over both depth and hidden-dimension for the Dense Reasoner. The experiments below use DeepSeek-R1 + ReasonIR-8B as the example pair (here d=4096). We can observe that:
> > >
> > > - Depth matters more than width: Going from 1→2→3 layers yields steady gains (1-layer is too shallow), but 5 layers degrad sharply, with unnecessary parameters and harder optimization,
> > >
> > > - Width has minor effect: Changing hidden dimension changes NDCG slightly.
> > >
> > > - Trade-off between performance and size: The 3-layer model gives the best nDCG but increases parameters by 33% relative to the 2-layer 0.73 nDCG gain. Given diminishing returns and inference-cost considerations, we select the 2-layer MLP with hidden size $d$ as the final design, which achieves near-top performance while keeping parameter count and inference cost low (and still negligible compared to any LLM call).
> > >
> > > We have included these tables and the parameter-cost analysis in the revised manuscript.
> > >
> > > | Layer | 1     | 2     | 3     | 4     | 5     |
> > > |-------|-------|-------|-------|-------|-------|
> > > | NDCG  | 23.04 | 23.83 | 24.56 | 24.13 | 17.33 |
> > >
> > > | Dimension | 2048  | 4096  | 6144  | 8192  |
> > > |-----------|-------|-------|-------|-------|
> > > | NDCG      | 23.47 | 23.83 | 23.77 | 23.92 |
> > >
> > >
> > >
> > > **2. Clarification of DIVER's results**
> > >
> > > We guarantee the accuracy of the results and provide the following clarifications, as shown in the table below:
> > >
> > > - Our reported DIVER (30% test) results refer specifically to the BRIGHT 30% test split used in our experiments (the split is stratified by source), so they are not directly comparable to the leaderboard numbers that report performance on the entire BRIGHT set.
> > >
> > > - We also ran a full reproduction on the entire BRIGHT set; those results are shown in the table as DIVER (reproduce) and align much more closely with the paper, presented as DIVER (paper). Additionally, the dense retriever we using is DIVER v1. The 31.9 you mentioned refers to DIVER v2's performance in the original query.
> > >
> > > | Methods           | Avg. | Bio. | Earth. | Econ. | Psy. | Rob. | Stack. | Sus. | Leet. | Pony | AoPS | TheoQ. | TheoT. |   |   |   |   |   |   |   |   |   |   |
> > > |-------------------|------|------|--------|-------|------|------|--------|------|-------|------|------|--------|--------|---|---|---|---|---|---|---|---|---|---|
> > > | DIVER (paper)     | 28.9 | 41.8 | 43.7   | 21.7  | 35.3 | 21.0   | 21.2   | 25.1 | 37.6  | 13.2 | 10.7 | 38.4   | 37.3   |   |   |   |   |   |   |   |   |   |   |
> > > | DIVER (reproduce) | 28.4 | 41.6 | 45.8   | 22.3  | 35.6 | 19   | 21.2   | 25.0   | 37.2  | 6.6  | 11.0   | 38.4   | 37.6   |   |   |   |   |   |   |   |   |   |   |
> > > | DIVER (30% test)  | 27.8 | 55.5 | 31.5   | 18.6  | 28.5 | 21.5 | 13.0     | 21.5 | 45.8  | 7.3  | 8.9  | 41.0     | 40.3   |   |   |   |   |   |   |   |   |   |   |

---

### Official Review · Reviewer_K9BT · 2025-11-04

**Soundness:** 2
**Presentation:** 2
**Contribution:** 2
**Rating:** 2
**Confidence:** 5

**Summary:**

The paper proposes an Adaptive Query Reasoning (AdaQR) framework that routes queries between an LLM based reasoning path vs a dense retriever trained to map an input query to an vector for an output query that represents a reasoned query.

The three main components are:
1. LLM Reasoner: The standard, high-cost approach of using an LLM to auto-regressively rewrite the query text.
2. Dense Reasoner: This is a lightweight model (a two-layer MLP) trained to approximate the effect of the LLM Reasoner. It operates directly in the embedding space, learning a transformation that maps an original query's embedding to an embedding that is close to the embedding of the LLM-rewritten query. This is based on the paper's key empirical finding that these embedding-space transformations are not random but "systematic [and] structured."
3. Reasoner Router: A lightweight routing mechanism that directs incoming queries. It compares a query's embedding to a pre-computed "oracle anchor" (an average of queries known to be handled well by the DR). Queries similar to this anchor are sent to the fast Dense Reasoner; all others are routed to the expensive LLM Reasoner.

Experiments on the reasoning-intensive BRIGHT benchmark, conducted across 5 retrievers and 17 LLM reasoners, show that AdaQR, on average, reduces the reasoning cost by 28.12% while simultaneously improving retrieval performance (nDCG@10) by 7.24%.

**Strengths:**

1. Provides a trainable solution for reasoning routing showcasing cost reduction on a variety of tasks
2. Showcases that reasoned queries are not random transforms of the input query

**Weaknesses:**

1. Misleading title. It is not a single model but a multi-stage pipeline that requires: (1) pre-training a Dense Reasoner on an external corpus, (2) fine-tuning the Dense Reasoner on an in-domain dataset, (3) building an "oracle anchor" for the router from the same in-domain data, and (4) deploying and maintaining both the lightweight DR and the full, expensive LLM Reasoner in production. How is the dense retriever a secret reasoner if it has been explicitly trained to learn the mapping?
2. The Reasoner Router is not self-learning; it relies on an "oracle anchor" built from a set of queries that are already known to be "predictable" by the Dense Reasoner. This requires an in-domain, labeled training set (in this case, 70% of the BRIGHT dataset) to construct the oracle, introducing a significant data dependency for the routing component. How will this generalize to other domains?
3. The routing mechanism is controlled by a similarity threshold that is "determine[d] empirically" and differs for each retriever backbone (e.g., 0.75 for BGE-Large, 0.6 for Qwen3-Embedding-0.6B). This is a sensitive, non-trivial hyper-parameter that requires manual tuning for any new deployment.
4. For custom deployments, the framework reduces the average reasoning cost but does not eliminate the peak deployment cost. The expensive LLM Reasoner must still be loaded in VRAM and available to serve "hard" queries, meaning the full hardware cost of the system is still incurred, even if the LLM is queried less frequently.

**Questions:**

na

---

> ### Author Response · Authors · 2025-11-20
> **Response to weaknesses**
>
> Thank you for the insightful comments. We have made the following response:
>
> **1. Meaning of secret reasoner**
>
> We agree that our original title was ambiguous. Our intention was to highlight that the Dense Retriever's embedding space provides a strong basis for capturing the reasoning-style mapping. The actual reasoning operation is performed by the Dense Reasoner, a small MLP trained to map original query embeddings to their LLM-rewritten counterparts. Calling it a "secret reasoner" meant that this lightweight model effectively performs the LLM's reasoning in embedding space without producing textual rewrites. We have clarified this distinction in the revised manuscript.
>
> **2. Generalization of AdaQR**
>
> We sincerely agree that the reviewer's insightful question regarding in-domain fine-tuning is critical. We omit the domain-specific fine-tuning stage for Dense Reasoner, relying solely on pre-training. During pre-training, we computed the router's oracle anchor instead of domain-specific anchor to validate AdaQR's generalization.
>
> Section 3.6 (Figure 7) shows detailed results of the ablation.
> Compared to AdaQR with domain-specific fine-tuning, there is a slight performance decline, with the average NDCG decreasing of 2.71%. However, compared to the LLM-rewritten NDCG 20.57, it still achieved a 4.53% improvement. Even without in-domain fine-tuning the Dense Reasoner, AdaQR still delivers strong gains over direct LLM-based rewriting, demonstrating that the method benefits are not solely driven by domain-specific adaptation. The full AdaQR pipeline does gain additional, measurable improvement from in-domain fine-tuning on BRIGHT's query style, but this improvement is not dominant. Overall, these results indicate good out-of-domain robustness and practical generalizability of AdaQR.
>
> **3. Hyper-parameter of Reasoner Router**
>
> We acknowledge the reviewer' s concern that the similarity threshold is a sensitive hyperparameter. In practice, Figure 6 shows that effective thresholds lie in a relatively narrow band (≈0.6 – 0.8) across retriever backbones, so the tuning burden is modest. Moreover, we implemented a simple automatic procedure that records the training-stage embedding distribution and selects a threshold by quantile (e.g., the p-th percentile of distances to the precomputed oracle anchor). In our experiments this quantile-based rule reliably reproduces manually tuned thresholds and substantially reduces manual effort.
>
> **4. The deployment cost of LLMs**
>
> We agree that peak hardware cost can remain if a large LLM must be kept available for hard queries. We would like to clarify two points.
>
> * In realistic production deployments many teams already provision LLM capacity (via API subscriptions or on-prem model servers) for other tasks. So maintaining an LLM is often not an incremental cost unique to retrieval.
>
> * The Dense Reasoner itself is extremely cheap (single MLP inference) and can be used without local LLM deployment, and even when used alone it yields 5% average nDCG improvement over out-of-the-box retrieval.
>
> Taken together, these facts mean AdaQR materially lowers average monetary cost and latency for reasoning-heavy retrieval while still providing a safe fallback to an LLM for hard cases.
>
> Thank you again for pushing us to think more deeply about the evaluation challenges in this domain.

---

> ### Author Response · Authors · 2025-11-26
> **We eagerly await your response!**
>
> We sincerely appreciate your time and effort in reviewing our manuscript and providing valuable feedback.
>
> We wish to confirm whether our responses have effectively addressed your concerns. We provided detailed responses to your concerns a few days ago and hope they have adequately resolved any issues. If you require further clarification or have any additional concerns, please do not hesitate to contact us. We are more than willing to continue our communication with you.
>
> Best regards.

---

### Comment · Area_Chair_EkDR · 2025-11-22

Dear Authors and Reviewers,

I would like to thank the authors for providing detailed rebuttal messages on time.

To reviewers: I would like to encourage you to carefully read all other reviews and the author responses and engage in an open exchange with the authors. Please post your first response as soon as possible within the discussion time window. Ideally, all reviewers will respond to the authors, so that the authors know their rebuttal has been read.

Best regards,
AC

---

### Note · Authors · 2025-12-01

I have read and agree with the venue's withdrawal policy on behalf of myself and my co-authors.